# Uncertainty-Aware Reinforcement Learning for Risk-Sensitive Player Evaluation in Sports Game

**Guiliang Liu**[1,2,3], **Yudong Luo**[1,3], **Oliver Schulte**[4], **Pascal Poupart**[1,3]

[1]University of Waterloo, [2]The Chinese University of Hong Kong, Shenzhen,
[3]Vector Institute, [4]Simon Fraser University

`liuguiliang@cuhk.edu.cn,yudong.luo@uwaterloo.ca,`
`oschulte@cs.sfu.ca,ppoupart@uwaterloo.ca`

## Abstract

A major task of sports analytics is player evaluation. Previous methods commonly measured the impact of players' actions on desirable outcomes (e.g., goals or winning) without considering the risk induced by stochastic game dynamics. In this paper, we design an uncertainty-aware Reinforcement Learning (RL) framework to learn a risk-sensitive player evaluation metric from stochastic game dynamics. To embed the risk of a player's movements into the distribution of action-values, we model their 1) *aleatoric uncertainty*, which represents the intrinsic stochasticity in a sports game, and 2) *epistemic uncertainty*, which is due to a model's insufficient knowledge regarding Out-of-Distribution (OoD) samples. We demonstrate how a distributional Bellman operator and a feature-space density model can capture these uncertainties. Based on such uncertainty estimation, we propose a Risk-sensitive Game Impact Metric (RiGIM) that measures players' performance over a season by conditioning on a specific confidence level. Empirical evaluation, based on over 9M play-by-play ice hockey and soccer events, shows that RiGIM correlates highly with standard success measures and has a consistent risk sensitivity.

## 1 Introduction

The advancement of player tracking and object detection systems enables data-driven analytics for professional sports players. A common approach to evaluating the contribution of players is to quantify their action impacts. Previous performance metrics [1, 2, 3, 4] computed the expected impact of an action on scoring or winning a game. However, actions with significantly different distributions of impact can have the same expectations. As a result, the expectation-based metrics cannot differentiate the risk-seeking actions from the risk-averse ones. How to distinguish these actions and assign proper credits to the players remains a fundamental challenge in sports analytics.

An important step toward a risk-sensitive evaluation metric is to model the distributions of action values. To achieve this goal, distributional Reinforcement Learning (RL) [5] predicts the supporting quantiles of action-value distributions. Previous distributional RL methods [6, 7, 8, 9, 10] mainly studied the virtual environments with deterministic transitions (e.g., Atari [11] or Mujoco [12]), whereas sports games are real environments with stochastic game dynamics and complex context features. Moreover, player evaluation, as a data-driven task, requires learning from a fixed dataset without exploration. The model must be able to handle the distribution shift experienced at test time.

To mitigate the impact of distribution shift, Offline RL algorithms [13] typically strive for conservative policies that discourage visits to Out-of-Distribution (OoD) states by penalizing corresponding values. However, this approach cannot be scaled to player evaluation since 1) our goal is not control (i.e., improving players' policy), but to use RL as an analytical tool to evaluate observed actions in professional games, and 2) penalizing actions in OoD states distorts the evaluation.

36th Conference on Neural Information Processing Systems (NeurIPS 2022).

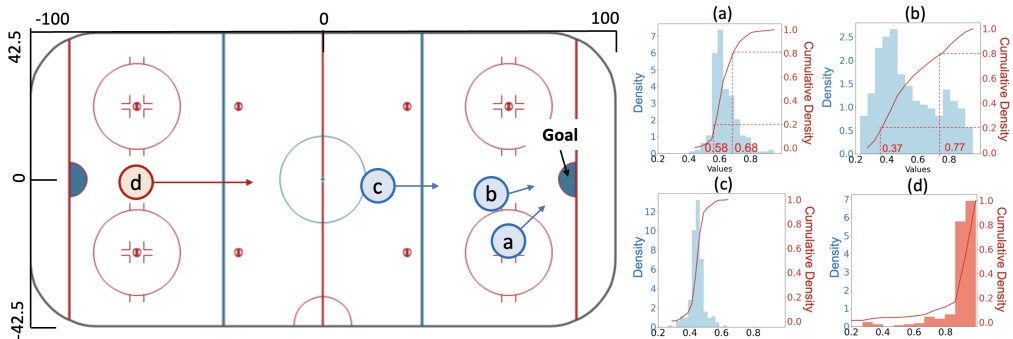

Figure 1: The predicted distribution of future goals in an ice hockey game between Blues and Coyotes, 2018-19 NHL season. The shots are made in the positions (a) - (d), providing important *motivations* for 1) **risk-sensitive evaluation**: Distributions (a) and (b) have *the same expectation* (around 0.6), but the first shot has a larger risk-averse estimate (at the confidence 0.8, we find $0.58 > 0.37$) and a smaller risk-seeking estimate (at the confidence 0.2, we find $0.68 < 0.77$), and thus they have *different impact on risk-sensitive evaluation*. 2) **Post-hoc calibration**: the event of shooting from the position (d) (the back-court) is rare in an ice hockey game, and thus this event is likely to be OoD, leading to a biased prediction at (d) (the predicted scoring chances are too large).

In this paper, we design an uncertainty-aware RL framework for risk-sensitive player evaluation. Figure 1 shows a real-world example introducing our key motivations. Instead of directly influencing players' actions like other RL algorithms, we perform a post-hoc calibration of the learned action values. The main idea of our framework is to model important types of uncertainty in sports games:

1) *Aleatoric uncertainty* captures the intrinsic stochasticity of game dynamics caused by stochastic rewards, transition dynamics, and policies. We show that this stochasticity can be captured by a distributional Bellman operator and propagated between action-value distributions by implementing Temporal-Difference (TD) learning with distributional RL.

2) *Epistemic uncertainty* is due to the finite training samples and OoD state-action pairs during testing. Online RL algorithms can overcome this uncertainty given sufficient exploration in the environment [8]. However, when we have only a demonstration dataset with limited samples, the influence of epistemic uncertainty cannot be ignored. Striving for simplicity and scalability, we model the epistemic uncertainty with a Feature-Space Conditional Normalizing Flow (FS-CNF).

Based on the uncertainty estimations, we develop a Risk-sensitive Game Impact Metric (RiGIM) for player evaluation. RiGIM filters the predictions for OoD samples and computes the impact of players' actions by conditioning on a confidence level. Empirical evaluation shows that RiGIM is highly correlated with standard measures when compared to other baselines. We measure the accuracy of action-value predictions by matching them empirically with game results and evaluate the risk-sensitivity of RiGIM by its correlations with standard measures at different confidence levels.

**Contributions.** 1) We design an uncertainty-aware RL framework that enables post-hoc calibrations on action values according to their aleatoric and epistemic uncertainties. 2) We demonstrate how the distributional Bellman operator captures the aleatoric uncertainty with action-value distributions from both a theoretical and an empirical perspective. 3) Striving for scalability, we design a feature-space density estimator that estimates the epistemic uncertainty with a minimum overhead. 4) To the best of our knowledge, RiGIM is the first risk-sensitive metric that incorporates the inherent risk in environment dynamics into player evaluation. Although this work mainly focus professional sports games, our method is general and can be scaled to other stochastic environments or domains.

## 2   Related Works

In this section, we introduce previous works that are most related to our approach.

**Uncertainty Estimation for RL.** Uncertainty estimates have been widely used in RL for guiding exploration and stabilizing policies. To achieve this goal, an effective approach is to measure the uncertainty of *future returns*:  [14] designed an uncertainty Bellman equation that estimates the

variance of the Q-value posterior distributions. Distributional RL methods [6, 7, 8, 9, 10, 15, 16, 17] directly model the distribution of future returns by computing corresponding quantities. Bootstrapped DQN methods [18, 19, 20, 21] learn ensembles of action-value Q functions to capture uncertainty. Some following works [22, 23] extend the Q-ensemble methods to offline RL settings by learning from a fixed dataset. Instead of focusing on the returns' uncertainty, an alternative approach is to measure the uncertainty of *model dynamics*: [24, 25] proposed model-based RL approaches that predict the uncertainty of dynamics models and penalize the actions leading to uncertain returns. Instead of separately capturing the epistemic and aleatoric uncertainties, these methods often quantify the overall uncertainty with a unified measure (e.g., variance or entropy).

Another line of approaches that heavily relies on uncertainty estimates is Risk-Sensitive RL (RSRL) [26]. The RSRL agents avoid states with high costs by estimating a risk measure (e.g., variance, Value at Risk (VaR), or Conditional VaR [27]). These algorithms [28, 29] often learn a controlling policy based on a known MDP instead of an offline dataset (with unknown dynamics).

**Player Evaluation.** The most common approach to player evaluation is to quantify the impact of their actions on game results [30]. Previous works measured action impacts by predicting 1) whether a goal will be scored within a fixed look-ahead horizon [3], 2) the change of winning chances [31], and 3) the expected number of goals within a possession [32]. Some recent works also trained action-value Q-functions by dynamic programming [1], deep Sarsa [2, 33] and Inverse RL [4]. These methods compute an expected action value without modeling their potential risk, and they commonly assume the training and testing datasets are identically distributed.

# 3 Uncertainty-Aware RL framework for Player Evaluation

We represent the dynamics in sports games with a Markov Game model and introduce the motivation of estimating the aleatoric uncertainty and the epistemic uncertainty.

## 3.1 Finite-Horizon Markov Game Model

Player evaluation metrics commonly evaluate players by how much their actions influence the opportunity of scoring the next goal [2, 3, 34]. Following this setting, we divide a sports game into *goal-scoring episodes*, so that each episode 1) begins immediately after a goal (or at the beginning of the game), and 2) terminates when the next goal is scored (or the end of the game is reached). From an algorithmic perspective, this setting allows us to bound the support of future-goals distribution (Section 4.1) into $[0, 1]$, which leads to faster model convergence and more accurate evaluations.

For a scoring episode of length $T_H$, we model its dynamics with a finite-horizon Markov game model [35]: $G = (\mathcal{S}, \mathcal{A}, P_{\mathcal{T}}, \mathbf{R}, \mathcal{O}, T_H, \gamma)$. At a time step $t \in [0, T_H]$, an agent $k$ performs an action $a_{k,t} \in \mathcal{A}_k$ at a game state $s_t \in \mathcal{S}$ after receiving an observation $o_t \in \mathcal{O}$. This process generates the next state $s_{t+1} \sim P_{\mathcal{T}}(\cdot|s_t, a_t)$ and a reward $r_{k,t} = R_k(s_t, a_{k,t})$. $\gamma$ is a discount factor. In this paper, we consider two agents $k \in \{Home, Away\}$ representing the home and away teams. The observed data $\mathcal{D} = [(o_1, a_{k,1}, \mathbf{r}_1), (o_2, a_{k,2}, \mathbf{r}_2), \ldots, (o_t, a_{k,t}, \mathbf{r}_t), \ldots]$ records the action $a_{k,t}$ performed by the team $k$ who possesses the puck. To alleviate the partial observability, a game state includes the game history: $s_t := (o_t, a_{t-1}, o_{t-1}, \ldots, o_0)$. The reward $\mathbf{r}_t$ is a 1-of-2 indicator vector that specifies which team ($Home, Away$) scores. We assign zeros to $\mathbf{r}_t$ until a team scores at the end of an episode.

## 3.2 Uncertainty-Aware RL for Player Evaluation

**Learning from Offline Data.** As a data-driven behavior analytic tool, the player evaluation model assigns values to players' actions by learning from an offline dataset. Under this setting, previous works [1, 2, 33, 3] commonly assumed the training and testing datasets are sampled from the same underlying distribution. However, in practice, since the behaviors of players may change when some team members (especially core players or the coach) are traded or hired during a season, there is no guarantee that the visitation frequency of state-action pairs is consistent in different games. It is natural to assume a distributional shift between the games in the training and testing dataset: while the value function is trained under one distribution, it will be evaluated on a different distribution.

**Calibration with Uncertainty.** To alleviate the influence of distribution shift, offline RL algorithms [13] commonly discourage the visit to OoD state-action pairs by lowering their values [36]

or penalising their rewards [24] or constraining the updated policy [37, 22]. However, to evaluate players' performance, RL is employed as a policy evaluation tool instead of controlling players. The trajectories in testing games correspond to observed players' actions that cannot be changed or influenced by penalties. Instead, we perform a post-hoc calibration of the predicted action values by modeling their *epistemic uncertainty*, which is due to a lack of knowledge about OoD samples, and thus the resulting model is uncertain about the returns (Section 4.2). Since our goal is to develop a risk-sensitive player evaluation metric, we estimate distributions of action values to model their *aleatoric uncertainty*, which is due to the intrinsic stochasticity in the game dynamics (Section 4.1). In practice, the quantification of aleatoric uncertainty can be influenced by the epistemic uncertainty of input samples, so we filter the OoD samples by utilizing our density estimator (Section 5.1) .

## 4 Modelling the Uncertainty of Action Values

The aforementioned uncertainty-aware RL framework requires estimating the aleatoric and epistemic uncertainty for a risk-sensitive player evaluation. In this section, we introduce our distributional-RL approach for modelling aleatoric uncertainty and a feature-space density estimator for measuring epistemic uncertainty (Figure 2 shows the model architecture).

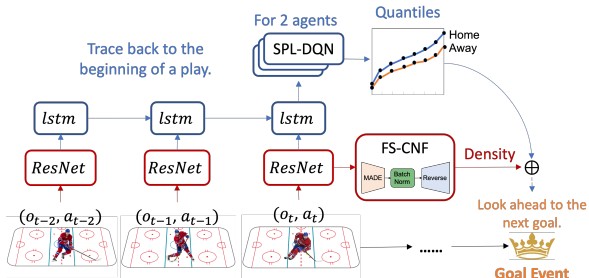
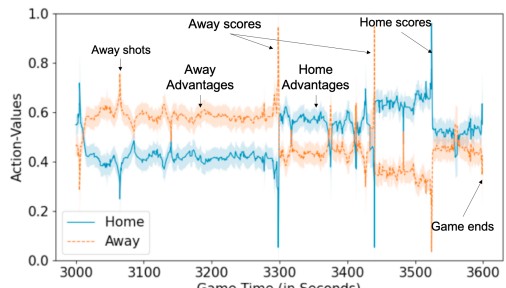

Figure 2: Model architecture. A play is a turn where one team attacks and the other defends. We add Spectral Normalization to ResNet outputs.

Figure 3: Illustrating the predicted distributions by showing the corresponding mean $\pm$ standard deviation at each time step in a sports game.

### 4.1 Distributional RL for Capturing Aleatoric Uncertainty

Distributional RL learns the distribution of the random variable $Z_k(s_t, a_t)$ that corresponds to the number of future goals when a player of team $k$ performs action $a_t$ in state $s_t$. In other words, we can think of $Z_k(s_t, a_t)$ as a random variable with outcomes corresponding to the sum of discounted rewards $\sum_{\iota=t}^{T_H} \gamma^\iota R_{k,\iota}(S_\iota, A_\iota)$, where $S_\iota = s_t$, $A_\iota = a_t$, $S_{\iota+1}$ is distributed according to $P_{\mathcal{T}}(\cdot | S_\iota, A_\iota)$ and $A_\iota$ is distributed according to $\pi(\cdot | S_\iota)$. Following the Quantile-Regression (QR)-DQN method [6], we represent the distribution of $Z$ by a uniform mixture of $N$ supporting quantiles by $\hat{Z}_k(s_t, a_t) = \frac{1}{N} \sum_{i=1}^{N} \delta_{\theta_{k,i}(s_t, a_t)}$, where $\theta_{k,i}$ estimates the quantile at the quantile level (or quantile index) $\hat{\tau}_i = \frac{\tau_{i-1} + \tau_i}{2}$ ($1 \leq i \leq N$, and $\tau_i = i/N$) and $\delta_{\theta_{k,i}}$ denotes a Dirac distribution at $\theta_{k,i}$. The model outputs $[\theta_{k,1}, \ldots, \theta_{k,N}]$ are monotonically increasing quantile values computed with the spline DQN (SPL-DQN) by following [17].

**Distributional Bellman Operator.** When the player of a team $k$ performs an action $a_t$ at a state $s_t$, the agent receives a reward $R_k(s_t, a_t)$ and moves to a future state $s_{t+1} \sim P_{\mathcal{T}}(S_{t+1} | s_t, a_t)$ where the agent's next action $a_{t+1} \sim \pi(A_{t+1} | S_{t+1})$. This stochastic process can be captured by a distributional Bellman operator $\mathcal{T}^\pi$ [6]:

$$\mathcal{T}^\pi Z_k(s_t, a_t) \stackrel{\Delta}{:=} R_k(s_t, a_t) + \gamma Z_k(S_{t+1}, A_{t+1}) \tag{1}$$

where $X \stackrel{\Delta}{:=} Y$ indicates that random variables $X$ and $Y$ follow the same distribution. Based on the distributional Bellman operator, we estimate the supporting quantiles of $Z$ by minimizing the quantile Huber loss (with threshold $\eta$):

$$\frac{1}{N} \sum_{i=1}^{N} \sum_{i'=1}^{N} \rho_{\hat{\tau}_i}^{\eta}(r + \gamma \theta_{i',k}(s_{t+1}, a_{t+1}) - \theta_{i,k}(s_t, a_t)) \text{ where}$$

$$\rho_\tau^\eta(\sigma) = |\tau - \mathbb{I}_{\sigma<0}|\mathcal{L}_\eta(\sigma) \text{ with } \mathcal{L}_\eta(\sigma) = \begin{cases} \frac{1}{2}\sigma^2, & |\sigma| \leq \eta \\ \eta(|\sigma| - \frac{1}{2}\eta), & \text{otherwise.} \end{cases} \quad (2)$$

*Illustration of Temporal Projection.* Figure 3 illustrates the mean $\pm$ standard deviation of the action-values sampled from the predicted distributions $\hat{Z}(s,a)$, where $s$ and $a$ follow the players' movements in a match between the Flyers (Home team) and the Maple Leafs (Away team) on March 15, 2019. The figure plots values of the two output nodes. We highlight critical events and match contexts to show the context-sensitivity of our predictions.

We show that the predicted distribution of action values can measure the aleatoric uncertainty.

**Proposition 1.** *Assume the Bellman consistency holds by $\hat{Z} \overset{\Delta}{:=} R + \gamma P^\pi \hat{Z}$ where $\hat{Z}$, $R$ are vector-valued random variables and $P^\pi$ is the transition matrix of the stationary policy $\pi$, so $P_{(s,a),(s',a')}^\pi = P(s'|a,s)\pi(a'|s')$, the uncertainty of action-value distributions $\hat{Z}$ under an entropy measure $H(\cdot)$ can be given by:*

$$H(\hat{Z}) = H[R] - |\mathcal{A}||\mathcal{S}|\log(1-\gamma) + \log|det(\mathbf{d}^\pi)| \quad (3)$$

*where $\mathbf{d}^\pi = (1-\gamma)(I - \gamma P^\pi)^{-1} \in [0,1]^{|S||A|\times|S||A|}$ is the induced matrix for distributions over state-action tuples by following policy $\pi$ and transition $P_\mathcal{T}$.*

The proof is in Appendix B.1. Proposition 1 disentangles the entropy of $Z$ into 1) the entropy of reward variables that quantifies the uncertainty of current rewards, 2) the uncertainty induced by the discount factor, which determines how much the current uncertainty estimation should be influenced by the stochasticity of future rewards or transitions (i.e., a small $\gamma$ reduces this influence), and 3) a log-absolute determinant of the induced distribution matrix, which measures the amount of stretch or change that the transition function $P_\mathcal{T}$ and the policy $\pi$ apply to the initial state-action distribution.

Proposition 1 demonstrates that the key components for representing the aleatoric uncertainty can be captured by $Z$ when the Bellman consistency is reached by learning, which suggests the action-value distribution learned by distributional RL is an ideal estimator for aleatoric uncertainty. However, in practice, the estimation of $Z$ cannot be well generalized to all samples because of insufficient exploration or limited training data, so we need to estimate their epistemic uncertainty.

### 4.2 Density Estimator for Capturing Epistemic Uncertainty

By definition, epistemic uncertainty stems from limited training data and is inherent to the model fitting these data. A common measure of epistemic uncertainty is $I(\theta; y|x, \mathcal{D})$ [38, 39, 40]: the amount of information gained when the model $\theta$ observes the true label $y$ of an input $x$. To estimate this uncertainty measure, previous works [41, 42, 43] utilized deep ensemble models and treated each ensemble as a sample from the posterior $p(\theta|\mathcal{D})$. However, adding additional deep ensemble layers to a distributional RL model (i.e., as a *unified* estimator for the joint distribution $p(Z_{1,...,K}, \Theta|\mathcal{D})$) significantly increases the model complexity. Striving for model simplicity and scalability for large datasets, we build a feature space density estimator [40] to detect OoD samples.

In this work, we design a Feature Space Conditional Normalizing Flow (FS-CNF) to estimate sample density in the training distribution with a minimum overhead. The main components are:

**Feature Extractor.** FS-CNF shares the same feature extracting layers with the distributional RL models. Note that a common reason why traditional feature extractors might fail to capture epistemic uncertainty is feature collapse [44], which maps OoD samples to iD regions in feature space. To prevent this phenomenon, an ideal feature extractor $f_\theta$ must be subjected to a bi-Lipschitz constraint:

$$\beta_1\|x_1 - x_2\|_I \geq \|f_\theta(x_1) - f_\theta(x_2)\|_F \geq \beta_2\|x_1 - x_2\|_I \; \forall x_1, x_2 \in \mathcal{D} \quad (4)$$

where 1) the lower Lipschitz bound ensures sensitivity to distances in the input space (i.e., sensitive to OoD samples) and 2) the upper Lipschitz bound ensures smoothness in the feature space (i.e., prevents overfitting to the input variations). To ensure this bi-Lipschitz condition in practice, we follow [44] and use residual networks with spectral normalisation as our feature extractor.

**Density Estimator.** Based on the extracted features, FS-CNF utilizes the Masked Auto-regressive Flow (MAF) [45] design that estimates the density of input variables in the training data distribution with an auto-regressive constraint. Additionally, to make FS-CNF more sensitive to the abnormal

predictions in OoD samples, the density model is conditioned on the expected returns of action-value distributions, so $p(\boldsymbol{x}|\boldsymbol{z}_E) = \sum_i p(x_i|x_{1:i-1}, \boldsymbol{z}_E)$ where $\boldsymbol{z}_E := \{\mathbb{E}[Z_k(s,a)]\}_{k=1}^K$ and $\boldsymbol{x} := (o, a)$ (we use $o$ instead of $s$ since a large input dimension influences the accuracy of estimation). We implement $p(x_i|x_{1:i-1}, \boldsymbol{z}_E) = \mathcal{N}(x_i|\mu_i, (\exp(\alpha_i))^2)$ where $\mu_i = \psi_{\mu_i}(x_{1:i-1}, \boldsymbol{z}_E)$ and $\alpha_i = \psi_{\alpha_i}(x_{1:i-1}, \boldsymbol{z}_E)$. The neural function $\psi$ is implemented by stacking multiple MADE layers [46].

## 5 Player Evaluation

In this section, we introduce our player evaluation metric and risk-sensitive rankings.

### 5.1 Risk-sensitive Impact Metric

**Measuring Risk.** We measure the risk of a player's action with the aleatoric uncertainty since modeling the intrinsic stochasticity of the game dynamics is consistent with the goal of sports analytics. However, as Figure 4 shows, when we quantify the aleatoric uncertainty with predictions from distributional RL, the performance is influenced by the density of input samples (i.e., OoD samples have a lower accuracy).

An effective approach to verify whether $\hat{Z}_k(s,a)$ accurately captures the true aleatoric uncertainty is to check the epistemic uncertainty for each input data $(s,a)$ [40]: 1) a high input density $(p(\cdot|\boldsymbol{z}_E) \geq \epsilon)$ indicates low epistemic uncertainty, (i.e., $(s,a)$ is inD) and we can trust the aleatoric uncertainty estimated by distributional RL. 2) a low input density $(p(\cdot|\boldsymbol{z}_E) < \epsilon)$ indi-

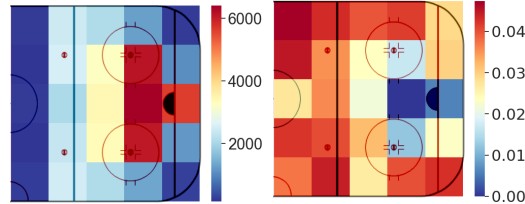

Figure 4: We discretize the ice hockey rink into 5×5 regions. For each region, the *left* heatmap shows the number of shots in the *training dataset*, and the *right* heatmap shows the Mean Absolute Error (MAE) between the estimated and the real aleatoric uncertainty for shot values in the *testing dataset*. We observe a negative correlation (-0.761) between the density and the MAE across these regions.

cates high epistemic uncertainty (i.e., the input is OoD, and we do not use the model prediction). In practice, $\epsilon$ is determined based on the validation dataset.

**Risk-Sensitive Action Impact.** The *impact* $\phi(s,a)$ measures how much an action $a$ changes the future return of a player's team. In terms of the value function, this is the change in action value due to a player's action. Previous works [1, 2, 3] computed the impact with the expected next-goal return $Q(s_t, a_t) = \mathbb{E}[Z(s_t, a_t)]$. $Q(\cdot)$ does not take into account the inherent variability of the returns and thus cannot estimate the risk of an action. To understand how players respond to risk, we propose a Risk-sensitive Game Impact Metric (RiGIM) based on $\hat{Z}_k(s,a)$ and $p(\cdot|\boldsymbol{z}_E)$:

$$\phi_k(s_{t+1}, a_{t+1}, c) = \left[\hat{Z}_k^c(s_{t+1}, a_{t+1}) - \hat{Z}_k^c(s_t, a_t)\right]\mathbb{I}_{p(\cdot|\boldsymbol{z}_E) \geq \epsilon}$$
$$RiGIM_l(c) = \sum_{(s,a) \in \mathcal{D}'} n(s,a,l) \times \phi_k(s,a,c) \tag{5}$$

where $c \in [0,1]$ is the confidence level, $Z^c$ denotes the $(1-c)^{th}$ quantile in $Z(\cdot)$ and $n(s,a,l)$ denotes the number of times that a player $l$ performs action $a$ at a state $s$ in the testing dataset $\mathcal{D}'$. We omit the term $r_t$ since $r_t = 0$ except at scoring step $T$, and $s_T$ is the terminal state of a scoring episode. $\phi(\cdot)$ can either be 1) risk-averse (with a large $c$), with better sensitivity to bad outcomes or 2) risk-seeking (with a small $c$), with better sensitivity to positive outcomes.

### 5.2 Case Study: Player Ranking in Testing Games

We rank players according to their RiGIM scores in the NHL testing games (see the experiment setting in Section 6) at different confidence levels. Tables 1 and 2 illustrates how a domain expert could use our method to gain insight into which types of players exhibit different risk-taking behavior. Table 1 shows a *risk-seeking* ranking (confidence $c = 0.2$), which favors offensive players (e.g., Centres (C)) with strong scoring ability. Aleksander Barkov, who scores the most points in these games, is captured by this ranking. When we set confidence $c$ to 0.8, Table 2 shows a *risk-averse* ranking which highlights players in defensive positions (e.g., Defensive (D)). John Klingberg, the

defenseman with the most assists, is listed in the top-10 players. We believe the differences between Tables 1 and 2 can be explained by the fact that RiGIM is correlated with the types of actions a player performs. Intuitively, in ice-hockey, some actions are more risk-seeking (shot) in terms of scoring while other actions are risk-averse (carry or pass). In general, players in the backcourt (e.g., defensemen) are more likely to perform risk-averse actions that have smaller variance on the scoring chances.



Table 1: Top 10 players with confidence 0.2.

| Player Name | Position | Team | P | A | G | RiGIM |
|---|---|---|---|---|---|---|
| Jonathan Toews | C | CHI | 10 | 5 | 5 | 14.72 |
| Anze Kopitar | C | LAK | 12 | 9 | 3 | 14.55 |
| Vincent Trocheck | C | FLA | 8 | 5 | 3 | 14.02 |
| Tomas Hertl | C | SJS | 12 | 8 | 4 | 13.97 |
| John Tavares | C | TOR | 12 | 3 | 9 | 13.92 |
| Tyler Seguin | C | DAL | 18 | 12 | 6 | 13.71 |
| Leon Draisaitl | C | EDM | 16 | 8 | 8 | 13.16 |
| Aleksander Barkov | C | FLA | 19 | 14 | 5 | 12.63 |
| Sean Couturier | C | PHI | 11 | 6 | 5 | 12.62 |
| Nathan MacKinnon | C | COL | 12 | 6 | 6 | 12.48 |

Table 2: Top 10 players with confidence 0.8.

| Player Name | Position | Team | P | A | G | RiGIM |
|---|---|---|---|---|---|---|
| Radek Faksa | C | DAL | 6 | 3 | 3 | 2.74 |
| Leon Draisaitl | C | EDM | 16 | 8 | 8 | 2.51 |
| John Klingberg | D | DAL | 10 | 9 | 1 | 2.46 |
| Esa Lindell | D | DAL | 3 | 1 | 2 | 2.29 |
| Connor McDavid | C | EDM | 18 | 11 | 7 | 2.23 |
| Tomas Hertl | C | SJS | 12 | 8 | 4 | 1.93 |
| Miro Heiskanen | D | DAL | 5 | 3 | 2 | 1.86 |
| Elias Pettersson | C | VAN | 8 | 6 | 2 | 1.79 |
| Tyler Seguin | C | DAL | 18 | 12 | 6 | 1.78 |
| Roope Hintz | LW | DAL | 11 | 7 | 4 | 1.77 |



# 6 Empirical Evaluation

**Dataset.** Our experiments utilize both a *ice-hockey* and a *soccer* dataset from the National Hockey League (NHL) and major European soccer leagues, which contain 9,213,371 events, covering 195 teams, 4,172 games, and 6,513 players. These datasets consist of events around the ball. Each event records the identity and action of the player possessing the ball, with time stamps and features of the game context (see all the game features in Appendix A.1). To the best of our knowledge, this is the *most extensive study for player evaluation*. Note that we *do not utilize virtual environments* like Atari [11] or Mujoco [12] because 1) the dynamics in these environments are deterministic without uncertainty to be modeled and 2) this paper mainly studies sports games that have stochastic dynamics [30], which are valid test-beds for our method.

**Experiment Settings.** We divide the dataset into a training set (80%), a validation set (10%), and a testing set (10%) according to game dates, so that games in the testing set happened after the games in the training and validation set. To predict the action values in the testing games, the metric must remain robust to OoD data points. We report the results averaged over 5 independent runs.

**Comparison Methods.** We employ an ablation design that iteratively removes parts from RiGIM . **GIM** removes the uncertainty estimator by directly using a Deep Recurrent Q-Network for estimating action values [2]. **T0-GIM** removes the recurrent model and uses a Deep Q-Network (DQN) for the value function. We then replace the RL framework with a supervised learning framework for estimating action values by following **VAEP** [3]. Instead of using function approximators, **Scoring Impact (SI)** [1] implements the tabular-based value iteration algorithm for computing action values from discretized spatial and temporal features. **Expected-Goal (EG)** metric directly uses the expected action values instead of impact values for measuring player performance. The last metric **Plus-Minus** $(+/-)$ is based on game statistics and measures the goal-gain with and without the player on court. We summarize these metrics in Table 3.

To study how well SP-CNF boosts model performance, we compare 1) a Gaussian Discriminant Analysis **(GDA)-RiGIM** metric that replaces SP-CNF with GDA [40], and 2) a Naive **(Na)-RiGIM** metric that removes the epistemic uncertainty estimator and uses all the predicted distributions to compute players' impact (see Appendix A.2 and A.3 for more details.).

Table 3: A summary of the baseline Methods for player evaluation.

| Method | Risk-Aware | History-Aware | RL-Based | Continuous Feature | Impact-Based | Context-Aware |
|---|---|---|---|---|---|---|
| $+/-$ | ✗ | ✗ | ✗ | ✗ | ✗ | ✗ |
| EG | ✗ | ✗ | ✗ | ✗ | ✗ | ✓ |
| SI | ✗ | ✗ | ✗ | ✗ | ✓ | ✓ |
| VAEP | ✗ | ✓ | ✗ | ✓ | ✓ | ✓ |
| T0-GIM | ✗ | ✗ | ✓ | ✓ | ✓ | ✓ |
| GIM | ✗ | ✓ | ✓ | ✓ | ✓ | ✓ |

## 6.1 Player Evaluation Performance: Correlations with Standard Measures

We follow [2, 4] and compute the correlations between player ranking metrics and standard measures on the testing games in a game season, because 1) the player (or agent) evaluation task has no ground-truth labels or rewards to maximize, 2) the correlation to all measures (including penalty measures) can measure whether the metrics can form a comprehensive evaluation to a player's overall

performance. We study 11 *success* measures for ice hockey and 8 *success* measures for soccer. To make the results more comprehensive, we also add 5 *penalty measures*. The studied measures are popular measures from the NHL and soccer statistics websites[1]. Following the popular risk-measures like Conditional Value at Risk (CVaR), we treat $c$ as a hyper-parameter. Since the comparison methods are expectation-based metrics, we study the options of 1) setting the confidence level ($c$) of RiGIM to 0.5 for a fair comparison 2) empirically determining $c$ with the validation set ($c^* = 0.34$ and $0.49$ for the ice hockey and soccer datasets). The risk-sensitive results are shown in Section 6.2.

Tables 4 and 5 show the average correlations in 5 independent runs on the testing dataset (see Tables C.1 and C.2 in Appendix for the complete mean $\pm$ standard deviation results). RiGIM achieves the highest correlations with 14 out of 19 success measures and the smallest correlations with 3 out of 5 penalty measures in ice hockey and soccer games. This observation shows that RiGIM is a comprehensive metric that can detect both the positive and the negative influence of a player. If we remove SP-CNF or replace it with other uncertainty estimators, most correlations become weaker except for the correlations with the SHP and SHG measures. This is because SHP and SHG rarely happen in a season (scoring with fewer players on ice is difficult). SP-CNF detects this phenomenon and assigns small densities to these rare events. RiGIM filters the event with a small density (see Equation 5), which might cause the loss of information and make its correlation with SHP and SHG less significant. SI correlates well with goal measures (Goals and Game Winning Goals) but has relatively poor correlations with other measures. This is because assigning an adequate value for *all* actions, including those with only intermediate effects on goal scoring, requires credit propagation over longer sequences, where neural nets are better at credit propagation than discretizing and using a tabular representation. For other risk-neutral methods, their performance is generally less satisfying when compared with risk-aware methods, especially for the $+/-$ metric, which shows the importance of capturing risk and modeling the context features.

Table 4: Correlations with standard measures in the **ice hockey** games. The *success* measures are assist, goal, Game Winning Goal (GWG), Overtime Goal (OTG), Short-handed Goal (SHG), Power-play Goal (PPG), Point (P), Short-handed Point (SHP), Power-play Point (PPP), Time On Ice (TOI), and Shots (S). The *penalty* measure is Penalty Minute (PIM).

| Methods | Assist | Goal | GWG | OTG | SHG | PPG | Point | SHP | PPP | TOI | S | PIM |
|---|---|---|---|---|---|---|---|---|---|---|---|---|
| $+/-$ | 0.181 | 0.189 | 0.187 | 0.028 | 0.071 | 0.063 | 0.206 | 0.119 | -0.071 | 0.021 | 0.038 | -0.014 |
| EG | 0.239 | 0.303 | 0.264 | 0.130 | -0.053 | 0.163 | 0.322 | 0.023 | 0.226 | 0.153 | 0.534 | -0.112 |
| SI | 0.237 | **0.596** | **0.409** | 0.123 | 0.095 | 0.351 | 0.452 | 0.066 | 0.274 | 0.224 | 0.405 | 0.138 |
| VAEP | 0.238 | 0.454 | 0.225 | 0.06 | 0.053 | 0.326 | 0.382 | -0.0 | 0.321 | 0.086 | 0.362 | 0.027 |
| T0-GIM | 0.397 | 0.394 | 0.139 | 0.16 | 0.151 | 0.216 | 0.455 | 0.153 | 0.295 | 0.356 | 0.387 | 0.058 |
| GIM | 0.456 | 0.408 | 0.167 | 0.158 | 0.134 | 0.246 | 0.501 | 0.137 | 0.345 | 0.395 | 0.431 | 0.061 |
| Na-RiGIM(0.5) | 0.593 | 0.476 | 0.223 | 0.173 | **0.152** | 0.313 | 0.625 | **0.175** | 0.453 | 0.597 | 0.611 | 0.115 |
| GDA-RiGIM(0.5) | 0.591 | 0.475 | 0.221 | 0.174 | **0.152** | 0.315 | 0.623 | 0.174 | 0.452 | 0.593 | 0.609 | 0.113 |
| RiGIM(0.5) | 0.675 | 0.477 | 0.266 | 0.184 | 0.11 | 0.355 | 0.678 | 0.141 | 0.529 | 0.68 | 0.7 | 0.146 |
| RiGIM($c^*$) | **0.68** | 0.477 | 0.269 | **0.187** | 0.107 | **0.357** | **0.681** | 0.141 | **0.531** | **0.685** | **0.707** | 0.147 |

Table 5: Correlations with standard measures in the **soccer** dataset. The *success* measures are goal, assist, Shots per Game (SpG), Pass Success percentage (PS%), Key Passes (KeyP), Dribbles (Drb), Crosses and (being) Fouled. The *penalty* measures are Yellow (Yel) and Red Card Received, Offsides (Off) and Own Goals (OwnG).

| Methods | Goal | Assist | SpG | PS% | KeyP | Drb | Crosses | Fouled | Yel | Red | Off | OwnG |
|---|---|---|---|---|---|---|---|---|---|---|---|---|
| $+/-$ | 0.284 | 0.318 | 0.199 | **0.288** | 0.218 | 0.119 | 0.017 | 0.035 | 0.001 | -0.069 | 0.053 | -0.001 |
| EG | 0.422 | 0.173 | 0.328 | 0.164 | 0.278 | 0.013 | 0.040 | -0.026 | 0.534 | 0.034 | -0.124 | -0.008 |
| SI | 0.585 | 0.153 | 0.438 | -0.140 | 0.052 | 0.050 | 0.216 | -0.065 | 0.114 | -0.089 | -0.249 | -0.102 |
| VAEP | 0.093 | 0.290 | 0.121 | -0.111 | 0.116 | 0.059 | 0.082 | -0.00 | 0.024 | 0.133 | -0.055 | -0.051 |
| T0-GIM | 0.614 | 0.455 | 0.715 | 0.148 | 0.472 | 0.431 | 0.161 | 0.355 | -0.007 | -0.027 | -0.346 | -0.168 |
| GIM | 0.627 | 0.462 | 0.72 | 0.149 | 0.473 | 0.437 | 0.169 | 0.358 | -0.0 | -0.025 | -0.336 | -0.154 |
| Na-RiGIM(0.5) | 0.646 | 0.507 | 0.741 | 0.144 | 0.503 | 0.445 | 0.177 | 0.391 | 0.101 | 0.007 | -0.309 | -0.144 |
| GDA-RiGIM(0.5) | 0.649 | 0.506 | 0.725 | 0.132 | 0.478 | 0.421 | 0.161 | 0.389 | 0.147 | 0.018 | -0.259 | -0.125 |
| RiGIM(0.5) | 0.671 | 0.577 | 0.756 | 0.181 | 0.574 | 0.530 | **0.239** | **0.448** | -0.092 | -0.039 | -0.451 | -0.185 |
| RiGIM($c^*$) | **0.682** | **0.583** | **0.757** | 0.186 | **0.575** | **0.531** | 0.238 | 0.446 | -0.101 | -0.042 | -0.455 | -0.184 |

---

[1] http://www.nhl.com/stats/skaters and https://www.whoscored.com/statistics

## 6.2 Sensitivity to Risk: Correlations Conditioning on Different Confidence Levels

We measure whether RiGIM is sensitive to the risk by its correlations with the standard measures, where RiGIM is conditioned on a specific confidence level $c$ (from 0 to 1), for example, RiGIM($c$), which indicates with probability $c$ that the players' impact should be at least RiGIM($c$).

Figures 5 and 6 show the correlations at different confidence levels for ice-hockey and soccer games. RiGIM is sensitive to risk, in the sense that it has different correlations with standard measures at these confidence levels, whereas GIM, as a risk-neutral metric, is unaware of the risk, and thus its correlations remain unchanged. Compared to other baselines, our RiGIM generally maintains higher correlations with success measures and lower correlations with penalty measures. The exceptions are the correlations with the SHP, SHG, and OTG. For the same reason as discussed above, SP-CNF may filter them during testing. We find when $c$ becomes smaller, RiGIM($c$) becomes risk-seeking, and thus achieves a higher correlation with success measures. However, the correlations drop when $c$ approaches 0. This is because $\hat{Z}_k^0$ denotes the estimates at the largest quantile level (see Equation 5), which corresponds to the most optimistic estimation action value (i.e., label scoring for all the shots). The overly risk-seeking estimation can induce a mismatch between estimated values and game facts, and thus cannot reflect the real contributions of players.

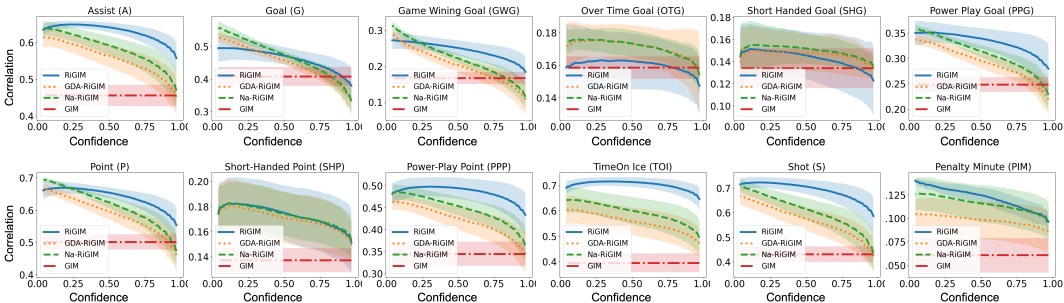

Figure 5: Correlations (Mean $\pm$ standard deviation) with success measures (the first 11 plots) and penalty measures (the last plot) at different confidence levels in **ice-hockey** games.

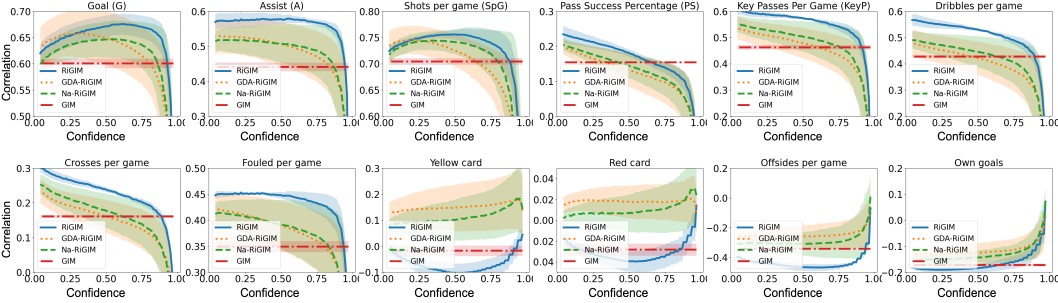

Figure 6: Correlations (Mean $\pm$ standard deviation) with success measures (the first 8 plots) and penalty measures (the last 4 plots) at different confidence levels for in **soccer** games.

## 6.3 The Prediction Accuracy: Match Future Scoring Frequencies With Action-Values

We study how well the predicted action-value distributions match the real next-goal scoring frequencies under discrete game contexts. These game contexts are constructed by dividing the continuous state space into discrete bins. To calculate the empirical scoring frequency associated with each bin, we assign an observed state $s$ to a bin $B$ according to the values of three discrete context features in the current observation: Manpower Differential, Goal Differential, and Period. The empirical and estimated scoring probabilities for a bin (with size $|B|$) are defined as follows:

a) *Empirical Scoring Chances*: for each $s \in B$, we set $g_k(s) = 1$ if the observed scoring-episode containing state $s$ ends with a goal by team $k$. Then $G_k^*(B) = \frac{1}{|B|} \sum_{s \in B} g_k(s)$.

b) *Estimated Scoring Chances*: for each $s \in B$, given $N$ samples from the calibrated distribution $z_k(s) \sim \hat{Z}_k(s,a)\mathbb{I}_{p(\cdot|\mathbf{z}_E)\geq\epsilon}$, the estimated chances are: $\hat{G}_k(B) = \frac{1}{N|B|}\sum_{s\in B}\sum_{n=1}^{N}z_{k,n}(s)$.

Table 6: The difference between the empirical and the estimated scoring chances in different contexts. Results are averaged over 5 runs $\pm$ standard error. $\uparrow$ ($\downarrow$) indicates that a difference is statistically greater (smaller) than the difference achieved by RiGIM with p-value $\leq 0.01$ according to the Wilcoxon signed rank test.

| | Ice-Hockey | | | Soccer | | |
|---|---|---|---|---|---|---|
| Manpower Differential | Short-Handed | Even-Strength | Power-Play | Short-Handed | Even-Strength | Power-Play |
| GIM | $0.115 \pm 0.078 \downarrow$ | $0.094 \pm 0.082 \downarrow$ | $0.099 \pm 0.085 \downarrow$ | $0.211 \pm 0.034 \downarrow$ | $\mathbf{0.114} \pm 0.05 \uparrow$ | $0.199 \pm 0.037 \downarrow$ |
| Na-RiGIM | $0.133 \pm 0.016 \downarrow$ | $0.064 \pm 0.016 \downarrow$ | $\mathbf{0.013} \pm 0.009 \uparrow$ | $0.226 \pm 0.019 \downarrow$ | $0.136 \pm 0.019$ | $0.175 \pm 0.028 \downarrow$ |
| GDA-RiGIM | $0.148 \pm 0.035 \downarrow$ | $0.072 \pm 0.029 \downarrow$ | $0.017 \pm 0.011 \uparrow$ | $0.216 \pm 0.022 \downarrow$ | $0.151 \pm 0.011 \downarrow$ | $0.18 \pm 0.013 \downarrow$ |
| RiGIM | $\mathbf{0.080} \pm 0.020$ | $\mathbf{0.058} \pm 0.008$ | $0.047 \pm 0.046$ | $\mathbf{0.204} \pm 0.005$ | $0.133 \pm 0.007$ | $\mathbf{0.147} \pm 0.033$ |
| Goal Differential | -1 | 0 | 1 | -1 | 0 | 1 |
| GIM | $0.238 \pm 0.122 \downarrow$ | $0.105 \pm 0.084 \downarrow$ | $0.271 \pm 0.059 \downarrow$ | $0.155 \pm 0.047 \downarrow$ | $0.155 \pm 0.054 \downarrow$ | $0.221 \pm 0.049 \downarrow$ |
| Na-RiGIM | $0.238 \pm 0.006 \downarrow$ | $0.045 \pm 0.015 \downarrow$ | $0.108 \pm 0.031 \downarrow$ | $0.157 \pm 0.02$ | $\mathbf{0.104} \pm 0.024$ | $0.16 \pm 0.017 \downarrow$ |
| GDA-RiGIM | $0.236 \pm 0.007 \downarrow$ | $0.045 \pm 0.016 \downarrow$ | $0.11 \pm 0.027$ | $0.165 \pm 0.018 \downarrow$ | $0.117 \pm 0.017 \downarrow$ | $0.175 \pm 0.007 \downarrow$ |
| RiGIM | $\mathbf{0.193} \pm 0.021$ | $\mathbf{0.029} \pm 0.015$ | $\mathbf{0.092} \pm 0.019$ | $\mathbf{0.152} \pm 0.008$ | $0.109 \pm 0.004$ | $\mathbf{0.149} \pm 0.013$ |
| Period | 3 | 2 | 1 | $2^{st}$ half | $1^{nd}$ half | N/A |
| GIM | $\mathbf{0.095} \pm 0.055 \uparrow$ | $0.111 \pm 0.086 \downarrow$ | $0.114 \pm 0.084 \downarrow$ | $\mathbf{0.191} \pm 0.037 \uparrow$ | $0.104 \pm 0.059 \downarrow$ | |
| Na-RiGIM | $0.139 \pm 0.018$ | $0.044 \pm 0.015 \downarrow$ | $0.024 \pm 0.015 \downarrow$ | $0.237 \pm 0.013 \downarrow$ | $0.061 \pm 0.03 \downarrow$ | |
| GDA-RiGIM | $0.143 \pm 0.028$ | $0.050 \pm 0.025 \downarrow$ | $0.033 \pm 0.025 \downarrow$ | $0.238 \pm 0.012 \downarrow$ | $0.059 \pm 0.026$ | |
| RiGIM | $0.143 \pm 0.011$ | $\mathbf{0.032} \pm 0.005$ | $\mathbf{0.014} \pm 0.009$ | $0.226 \pm 0.011$ | $\mathbf{0.058} \pm 0.009$ | |

Table 6 shows the average absolute difference between $\hat{G}$ and $G^*$ based on 5 independent runs. We implement a Wilcoxon signed rank test [47] to study whether the predictions from baseline methods are different from that of RiGIM for all samples. Our baselines are the learning-based action-values metrics for the context-aware evaluation. RiGIM achieves a minimum distance in 12 out of 17 bins. This is because 1) FS-CNF outperforms GDA by computing a more accurate uncertainty estimator for filtering OoD states and 2) the distribution estimates contain richer information than expectation estimates, allowing $\tilde{Z}(\cdot)$ to better match the scoring frequencies than $Q(\cdot)$. This observation is consistent with the findings in [6].

## 6.4 Limitations

**A Study for Sports Games.** This work uses stochastic sports games as the testbeds for uncertainty estimation, but we argue the same methods can be easily migrated to other applications in stochastic environments such as autonomous driving or healthcare. Our approach of measuring aleatoric and epistemic risks can be adapted to offline RL for learning conservative policies.

**Evaluation Instead of Control.** Our method focuses on player evaluation instead of control. We believe both tasks are challenging with respect to different aspects. Evaluation requires the action values to accurately reflects an agent's real contribution to game-winning. An ideal evaluation metric can provide in-game predictions of game outcomes, which is important for the sports industry.

## 7 Conclusion and Future Work

In this paper, we designed an RL framework for quantifying the aleatoric uncertainty and the epistemic uncertainty from stochastic sports datasets. This framework enabled distributional RL and an FS-CNF model to estimate both uncertainties, with which we proposed a risk-sensitive evaluation metric RiGIM . Empirical results show that RiGIM correlates well with success measures and the correlation is sensitive to different risks. A direction of future work is to extend our model to other domains.

## Acknowledgments and Disclosure of Funding

We acknowledge the funding from the Canada CIFAR AI Chairs program, and the support of the Natural Sciences and Engineering Research Council of Canada (NSERC). Resources used in this work were provided, in part, by the Province of Ontario, the Government of Canada through CIFAR, and companies sponsoring the Vector Institute https://vectorinstitute.ai/partners/.

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

# 8 Submission of papers to NeurIPS 2022

