# OpenReview forum: "Uncertainty-Aware Reinforcement Learning for Risk-Sensitive Player Evaluation in Sports Game"
_NeurIPS.cc/2022/Conference — NeurIPS 2022 Accept_

### Official Review · Reviewer_mdqR · 2022-06-26

**Rating:** 7
**Confidence:** 3
**Ethics Flag:** Yes
**Soundness:** 3 good
**Presentation:** 3 good
**Contribution:** 2 fair

**Summary:**

The paper addresses the problem of evaluating player behavior in games based on the expected value and uncertainty of actions taken, specifically when examining data obtained from live sports (ice hockey and soccer).

The authors propose a model to decompose games into a series of episodes with inherent stochasticity ("aleatory uncertainty") and limited observational data ("epistemic uncertainty"). The model uses distributional reinforcement learning to address stochastic dynamics. The model also filters samples using a density estimator to address out-of-distribution data and offline data sampling sparsity. Evaluation results show better correlations to game metrics (like scoring goals) when compared to alternative accepted methods and small errors when predicting scoring probabilities in different conditions.

**Questions:**

**[Q0]** What does employing a risk-based cutoff add to the existing analyst techniques? Having a variety of cut-off thresholds (like all hyperparameters) makes assessment more complex, begging the question: what are the benefits for these additional complexity costs? The empirical results make one useful case about better calibration, but the text would benefit from clarifying the impact and significance to how analysts (or players or coaches) would use this new information.

**[Q1]** What new insights can be obtained from the rankings provided? How are these of interest to those with domain expertise? Practitioners?

**[Q2]** How does RiGIM scale? How much does the model improve as the number of matches observed increases? How effective is the model for evaluating new players (presumably since the player is not a feature of the model cold start problems should be lesser)? How does model training scale in terms of compute resources needed?

**[Q3]** (Table 4) Why might SI do so well at predicting Goals and Game Winning Goals compared to the alternatives? There is a substantial gap between the performance of SI and the runner up for Goals (0.596 vs 0.477) and GWG (0.409 vs 0.266). What is RiGIM failing to capture that SI does?

**Ethics Review Area:**

["Privacy and Security (e.g., consent)"]

**Limitations:**

In terms of limitations, it would help to read some of the authors thoughts on scalability of this technique (mentioned above in the questions). What aspects of risk remain poorly disentangled that are worth further investigation?

The discussion of the negative impacts of increased player scrutiny is welcome in an era where greater analytics empowers audiences to (often harshly) critique players. I would suggest discussing some of the effects of sports analytics on the behavior of coaches (team decision-makers more broadly) when it comes to evaluating and understanding their players. New analytics can provide insight, but there is always the risk of Campbell's Law and increasing pressure on players to conform to metrics over good behavior (not new in the sense that scoring behavior is already tracked, but now with new measures).

**Strengths And Weaknesses:**

# strengths

The method improves over other methods. The results for predict scoring chances are particularly encouraging. The correlational results are more mixed, though the family of proposed methods (*-RiGIM) as a whole obtain competitive results. The bootstrap repetitions in the appendix are convincing that the results are not mere sampling artifacts (though would be welcome in the main text).

The paper's originality lies in quantifying risk taking (seeking) behavior from offline and potentially sparse data. The core techniques are not particularly novel (distributional RL & density estimation), but they are clearly explained and integrated into a relevant application context. Many analytics practitioners would benefit from understanding these methods and considering the use of uncertainty estimates in their algorithms.

The exposition is clear and generally easy to follow. Some of the proofs require further background knowledge (not surprising) and the dataset was not clearly explained in the main text. These are minor flaws and less important than conveying the core approach and goals.

The method shows promising generality. Evaluations on both ice hockey and soccer are favorable, suggesting the decomposition can impact a variety of sports analytic fields (among other domains). This is a welcome advance given the increasing adoption of analytics in many sports and the diversity of existing practices for making human judgments of player performance. Generalization to other offline RL tasks remains an open question, but suggests a broader potential audience at NeurIPS.

The evaluations demonstrate rigor in handling real-world offline data. For example, splitting time periods and ensuring test data comes later, as would occur when deploying this approach in real world applications.


# weaknesses

The RiGIM model results (Figure 5) are ambiguous. The error bars for ablations heavily overlap the full model, suggesting that the simpler Na-RiGIM may be sufficient in most contexts. The text would benefit from addressing the differences in performance and why they may (not) obtain in this case.

Limited technical novelty. Distributional Bellman and the feature density estimator are both adapted from prior work. The novelty lies more in application than new major results. This is a minor weakness: combining the methods is not trivial.

It is not clear why risk-sensitive evaluation is valuable to practitioners. Given the post-hoc nature of the analysis, risk quantification will not help balancing exploration and exploitation. This opens the question as to how significant the results are to practitioners. The text would benefit from explicitly addressing this point (see question below).

---

> ### Author Response · Authors · 2022-08-01
> **Author response to comments from reviewer mdqR**
>
> Dear Reviewer, we appreciate your constructive comments. We have revised the paper with the following clarification as per your feedback,
>
> - "*What does employing a risk-based cutoff add to the existing analyst techniques?...*"
>
> **Response**: The cut-off, determined by confidence value $c$, is proposed to differentiate risk-seeking players from risk-averse ones. Following the designs of popular risk measures like Conditional Value at Risk (CVaR), we treat $c$ as a hyper-parameter.  In practice, we allow the coaches or managers to pick a confidence value depending on whether they want to find the risk-seeking players (with a small $c$) or the risk-averse ones (with a large $c$). Figure 1 shows an example: if we set $c=0.8$, the action (a) will have larger values and the corresponding player could be assigned more credits. However, when we set $c=0.2$, the player who performs the action (b) should be ranked higher, although these two actions (a and b) have the same expected value. We have added clarification to our paper (see lines 287-290).
>
> - "*What new insights can be obtained from the rankings provided? How are these of interest to those with domain expertise? Practitioners?*"
>
> **Response**: Table 1 shows a risk-averse ranking (with confidence 0.2) that favors offensive players (e.g., Centres (C)) with strong scoring ability. Table 2 shows a risk-sensitive ranking that highlights players in defensive positions. We use Aleksander Barkov and John Klingberg as two examples. These tables illustrate how a domain expert could use our method to gain insight into which types of players exhibit different risk-taking behavior. Also Figures 5 and 6 indicate which action types are riskier. In practice, teams and managers are encouraged to use our method for predictive estimation of players' performance.  We have significantly improved section 5.2.
>
> - "*How does RiGIM scale? and How does model training scale in terms of computing resources needed?*"
>
> **Response**: We believe RiGIM can be easily extended to offline RL and other learning-from-demonstration applications given a trajectory dataset (see Section 6.4 and Appendix A.1). In this work, since the inputs to the models are symbolic features (i.e., features with physical meaning like x-y coordinates and velocities), we simply use a fully-connected neural layer as the feature extractor. However, if the data is in the format of images and text, stronger feature extractors will be required. We report the Computational Resources and Running Time in Appendix A.4.
>
> - "*How much does the model improve as the number of matches observed increases?*"
>
> **Response**: RiGIM is uncertainty-aware. Intuitively, increasing the number of training games reduces the scale of epistemic uncertainty, but the scale of aleatoric uncertainty will not decrease ([8] explores this phenomenon), so our uncertainty-aware method should still out-perform risk-neural methods.
>
> - "*How effective is the model for evaluating new players (presumably since the player is not a feature of the model cold start problems should be lesser)?*"
>
> **Response**: The inputs of our model are state-action pairs without including the identities of players. Thanks to the generalization ability of neural models, we can assign a proper value to the movements of unknown players if his or her playing style is similar to that of players in the training dataset.
>
> - "*(Table 4) Why might SI do so well at predicting Goals and Game Winning Goals compared to the alternatives? There is a substantial gap between the performance of SI and the runner up for Goals (0.596 vs 0.477) and GWG (0.409 vs 0.266). What is RiGIM failing to capture that SI does?*"
>
> **Response**: Table 4 is for player evaluation instead of predicting goals. In other words, the model knows the scoring states (with reward 1)  during a game. The goal is not to predict whether a goal will be scored after performing an action in a state but to assign all the state-action pairs (mostly with reward 0) some proper credits. If the credits are correctly assigned, the corresponding metric should be well correlated with most of the well-known success measures. When it comes to SI, this is a model based on discretizing the continuous state features (time and space), then applying dynamic programming to a tabular state representation (See citation [1]).
>     It correlates well with goal measures (Goals and Game Winning Goals) but has relatively poor correlations with other measures. This is because assigning an adequate value for **all** actions, including those with only intermediate effects on goal scoring, requires credit propagation over longer sequences (up to 13 steps). For continuous spatio-temporal processes like ice hockey, neural nets are better at credit propagation than discretizing and using a tabular representation.
>     This finding is consistent with previous works [4] and [33]. We add the discussion to our paper (lines 301-205)

---

> > ### Comment · Reviewer_mdqR · 2022-08-04
> > **reviewer response**
> >
> > Thank you for the detailed responses!
> >
> > - _In practice, we allow the coaches or managers to pick a confidence value depending on whether they want to find the risk-seeking players (with a small $c$) or the risk-averse ones (with a large $c$)._
> > - _These tables illustrate how a domain expert could use our method to gain insight into which types of players exhibit different risk-taking behavior._
> >
> > These remarks regarding practitioner interpretation using risk-seeking or risk-averse choices of $c$ greatly clarify how to use these estimates in practice. The text is stronger for the additional detail on these points!
> >
> >
> > - _When it comes to SI, this is a model based on discretizing the continuous state features..._
> >
> > This is very helpful and adding it to the paper enables readers to better understand the contribution and application context.
> >
> > - _Thanks to the generalization ability of neural models, we can assign a proper value to the movements of unknown players if his or her playing style is similar to that of players in the training dataset._
> >
> > Adding a comment along these lines might strengthen the message in the paper. Regardless it addresses what I was asking about.
> >
> > ## Open questions
> >
> > - _We report the Computational Resources and Running Time in Appendix A.4._
> >
> > Apologies, this seems to be a point of miscommunication. My question around scalability was intended to ask about the big-$O$ scalability of runtime or memory of the algorithm. Or an empirical estimate of it such as running the model on 20%, 40%, 60%, 80%, 100% of the data and assessing the runtime and GPU memory needs.
> >
> > - _Intuitively, increasing the number of training games reduces the scale of epistemic uncertainty, but the scale of aleatoric uncertainty will not decrease ([8] explores this phenomenon), so our uncertainty-aware method should still out-perform risk-neural methods._
> >
> > Is there any empirical validation of this for the model (even if only the full RiGIM model)? Practitioners would presumably want some validation that the model with more data will better support their decision-making through more accurate assessments. I believe the scaling explanation, but given the task (separating aleatory & epistemic uncertainty) is novel it would help to demonstrate this holds empirically.

---

> > > ### Author Response · Authors · 2022-08-05
> > > **Thanks for your reply and more ideas about your concerns**
> > >
> > > Dear reviewer, we are greatly thankful for your reply. We hope the following response can solve your concerns.
> > >
> > >
> > > - "*My question around scalability was intended to ask about the big-O scalability of runtime or memory of the algorithm. Or an empirical estimate of it such as running the model on 20\%, 40\%, 60\%, 80\%, 100\% of the data and assessing the runtime and GPU memory needs.*"
> > >
> > > We apologize for our misunderstanding. We include an analysis of the computational complexity and memory complexity in appendix A.4.
> > >
> > > - "*Is there any empirical validation of this for the model (even if only the full RiGIM model)? Practitioners would presumably want some validation that the model with more data will better support their decision-making through more accurate assessments. I believe the scaling explanation, but given the task (separating aleatory & epistemic uncertainty) is novel it would help to demonstrate this holds empirically.*"
> > >
> > > We will add an experiment to validate our claim. As [8] shows, the scale of epistemic uncertainty will reduce as more data are observed. In our work, epistemic uncertainty is measured by the feature-space density estimator. We will add an experiment to compute the scale of the density of a testing game as more games are observed during training. We will report the results and notify the reviewer as soon as our experiment is done.

---

> > > > ### Comment · Reviewer_mdqR · 2022-08-07
> > > > **comment on scaling**
> > > >
> > > > Great! Looking forward to the experiment results.
> > > >
> > > > > memory complexity of RiGIM is $O[2I^2 + 4IH + (N + 1)H^2 + 3KIH + K(L − 1)H]$
> > > >
> > > > I think I missed something: why is batch size independent of memory needs?

---

> > > > > ### Author Response · Authors · 2022-08-09
> > > > > **New Results**
> > > > >
> > > > > Thanks for your patience!
> > > > >
> > > > > We have added the experiment results to Appendix C.4, where we illustrate the scale of epistemic uncertainty after observing more training games. The uncertainty is measured by the negative log-likelihood $-\log p(s,a|z)$ based on the outputs from the feature-space density estimator. The results provide evidence that data plays an important role in supporting decision-making. Hope it can resolve your concerns!
> > > > >
> > > > > In terms of the memory complexity, we follow previous works and mainly focus on the size of model parameters, which is independent to the batch size, but we agree that the batch size must be considered in practice. We have added it to our paper.

---

### Official Review · Reviewer_kCNJ · 2022-07-11

**Rating:** 6
**Confidence:** 3
**Soundness:** 3 good
**Presentation:** 4 excellent
**Contribution:** 3 good

**Summary:**

The authors develop an offline RL model for risk-sensitive agent evaluation and apply it to study player impact in sports games. Unlike previous approaches, their method RiGIM is sensitive to stochasticity in the environment (some situations have lower/higher variation in their outcomes) and can handle the OoD actions that plague Offline RL. Because they are focused on evaluation (and not control), the typical techniques for dealing with OoD actions (lowering action values or constraining the policy) will not work and so the authors develop a scalable density estimator based on normalizing flows. These techniques are evaluated on a large dataset of hockey and soccer games and correlated with traditional measures of impact.

**Questions:**

Can you give some intuitions for the case study results in Table 1/2. The only comment is that there are more defensemen in the higher confidence top 10 -- although this is true empirically -- why is this a prediction of higher confidence? It doesn't seem to clearly correlate with the other metrics shown. Tyler Seguin and Connor McDavid have similar stats and play in the same position so what accounts for them being in the different top 10 lists? The authors mention a couple of qualitative findings e.g., "low confidence favors centers, strong scoring ability) but its not clear what is driving these predictions. How would a layperson use this tool?

For tables 4/5/6, why was c set to 0.5 instead of fit on the validation set? How would one decide on a value for c in practice?

In table 6, many of the bolded numbers are not different at a level of statistical significance. Ideally, the algorithm can be run more times for the camera ready to reduce the uncertainty about performance.

Why is the correlation with traditional metrics a good way of measuring success? The "success measures" are very simplistic features of the game so it could be that a better correlation with these measures is actually a signal that the algorithm is doing something naive rather than sophisticated.

Why do traditional metrics better correlate with risk-seeking versions of (smaller values of c)?

Will the post-hoc calibration techniques developed in this work also apply to the offline RL control setting? Might this be a better approach than constraining the actions that the agent can take?

**Limitations:**

Limitations are addressed.

**Strengths And Weaknesses:**

The paper is clearly written with consistent notation and the problem is well articulated. Both the problem statement and the techniques used to solve it are novel to my knowledge. Furthermore the approach is demonstrated on a large dataset of practical importance with potentially promising results. The main weaknesses are related to the clarity and interpretation of the evaluation (expanded on in the Question section). The evaluation of the results are confusing and it is not clear how this technique should be measured or used.

Small issues:

Line 37-39 citation of what algorithms are being contrasted here.
What is the x-axis supposed to represent in (a)-(d) in Figure 1?

Grammar: "We measure whether RiGIM is sensitive to the risk by its correlation" (Line 287).

Lines 294-296 are repetitive with 282-283

---

> ### Author Response · Authors · 2022-08-01
> **Author response to comments from reviewer kCNJ -- Part1**
>
> Dear Reviewer, we appreciate your constructive comments. We have revised the paper with the following clarification as per your feedback,
>
> - "*Can you give some intuitions for the case study results in Table 1/2. ....The only comment is that there are more defensemen in the higher confidence top 10...why is this a prediction of higher confidence? what accounts for Tyler Seguin and Connor McDavid being in the different top 10 lists?...*"
>
> **Response:** Our paper shows that the risk-averse ranking includes more defensemen. This illustrates how a layperson could use our method to gain insight into which types of players exhibit different risk-taking behavior. Our intuitive explanation for this finding is that the RiGIM risk metric is correlated with the action types a player performs. Intuitively, in ice hockey, some actions are more risk-seeking (shot) in terms of scoring while other actions are risk-averse (carry or pass). In general, players on the backcourt (e.g., defensemen) are more likely to perform risk-averse actions that have smaller variance and are not directly related to shooting and scoring. In terms of the rankings for Tyler Seguin and Connor McDavid, stats are based only on goals and assists, whereas the RiGIM metric uses all the play-by-play data so it is context-aware and takes into account much more information. So we would not expect RiGIM to always agree with the conventional stats. We provide the stats to provide some context for readers who may not be familiar with who these players are. Also it shows that our ranking passes the ``eye test" in that well-known stars that are big goal scores/contributors stand out. We have expanded section 5.2 (see lines 243-247).
>
> - "*For tables 4/5/6, why was c set to 0.5 instead of fit on the validation set? How would one decide on a value for c in practice?*"
>
> **Response:** This is because the comparison methods are expectation-based metrics. For a fair comparison, in section 6.1, we set $c$ to 0.5 for a risk-neutral version of RiGIM. The risk-sensitive results are shown in Section 6.2. Setting $c$ to the value that maximizes the RiGIM correlations in the validation set could be an alternative approach. Intuitively, it will further improve the RiGIM performance, although the comparison could be a little biased. In practice, the coaches or managers should pick a confidence value depending on whether they want to find the risk-seeking players (with a small $c$) or the risk-averse ones (with a large $c$). This is similar to determining the confidence value in the Value-at-Risk (VaR) or Conditional VAR (CVaR) measures. We have added the explanation to our paper (lines 287-290).
>
> - "*In table 6, many of the bolded numbers are not different at a level of statistical significance. Ideally, the algorithm can be run more times for the camera ready to reduce the uncertainty about performance.*"
>
> **Response:** Table 6 reports MAE at each bin constructed by discretizing the game context. The differences between the estimated scoring chances and real scoring chances are averaged over all the state-action pairs in a bin. The difference is significant based on a Wilcoxon signed rank test and results for all samples, but the difference might be less significant if we look at the averaged numbers. We have shown these results in Table 6.
>
> - "*Why is the correlation with traditional metrics a good way of measuring success? ....*"
>
> **Response:** We study the correlation to success measures because 1) Unlike the supervised learning task or the RL controlling task, the player (or agent) evaluation task has no ground-truth labels or rewards to maximize, so we follow previous studies [2,4] and use the correlation. 2) In the experiment, we study the correlation to all measures (including some penalty measures) instead of one. In this way, we can know whether these player evaluation metrics can form a comprehensive evaluation of a player's overall performance. We have added these ideas to our paper (see lines 281-285).
>
> - "*Why do traditional metrics better correlate with risk-seeking versions of (smaller values of c)?*"
>
> **Response:** We assume the reviewer is asking for an explanation about the results of the risk-sensitive experiment in Section 6.2. In this experiment, we find that when c becomes smaller, RiGIM becomes risk-seeking, and thus achieves a higher correlation with success measures. Firstly, we want to clarify that the curve is not monotonically decreasing (see lines 296-298). The curve often reaches its maximum with a small $c$. This is because, in general, a risk-seeking metric assigns larger values to risk-seeking actions like shots. Compared to other actions, risk-seeking actions are more correlated to success measures like a goal. However, when $c=0$, the metric becomes overly risk-seeking and the correlation will drop. We have expanded the discussion (see lines 320-323).

---

> > ### Author Response · Authors · 2022-08-01
> > **Author response to comments from reviewer kCNJ -- Part2**
> >
> > - "* Will the post-hoc calibration techniques developed in this work also apply to the offline RL control setting? Might this be a better approach than constraining the actions that the agent can take?*"
> >
> > **Response:** Our work is based on some settings in Offline RL (see "learning from offline data" in Section 3.2). Although we solve an agent-evaluation task instead of a controlling task, we believe the approach of measuring aleatoric and epistemic risks can be adapted to Offline RL. With the risk measure, the offline RL agent can prevent state-action pairs with large uncertainty during testing.

---

> > ### Comment · Reviewer_kCNJ · 2022-08-05
> > **Reply**
> >
> > Thank you for the response. These comments and additions improve the paper but I have decided not to increase my score. I still support the publication of this manuscript.
> >
> > > Intuitively, in ice hockey, some actions are more risk-seeking (shot) in terms of scoring while other actions are risk-averse (carry or pass).
> >
> > Wouldn't these actions have different expected values and not just be different in terms of risk-seeking? I don't understand why a defenseman would be lower variance. Or is that an emergent finding of this study?
> >
> > > In terms of the rankings for Tyler Seguin and Connor McDavid, stats are based only on goals and assists, whereas the RiGIM metric uses all the play-by-play data so it is context-aware and takes into account much more information. So we would not expect RiGIM to always agree with the conventional stats.
> >
> > Does this reveal that their EV is just so high that they are included regardless of risk preferences? I feel that there is a real missed opportunity to explain what the metric is doing in terms that could be useful to a practitioner.
> >
> > > Compared to other actions, risk-seeking actions are more correlated to success measures like a goal.
> >
> > I think the missing explanation is that I'm having a hard time reasoning about what a risk-averse but highly successful action should look like. Everything that is positive seems to qualify as "risk-seeking" to some extent. Is there some way that this metric might lead to a more nuanced understanding of defensive behavior (which don't seem to have as strong traditional metrics associated with them).
> >
> > >  Intuitively, it will further improve the RiGIM performance, although the comparison could be a little biased.
> >
> > Why would it be biased? There is no reason to think that the conventional metrics represent a risk-neutral measure of play. If you tune the hyper-parameter on a validation set that is unlikely to introduce bias.

---

> > > ### Author Response · Authors · 2022-08-05
> > > **Thanks for your reply and more ideas about your concerns**
> > >
> > > Dear reviewer, we are greatly thankful for your reply. We hope the following response can solve your concerns.
> > >
> > > - "*Wouldn't these actions have different expected values and not just be different in terms of risk-seeking? I don't understand why a defenseman would be lower variance. Or is that an emergent finding of this study?*"
> > >
> > > Yes, it is true that the expected values of shots should be larger, but we also observe that the variance of the shot distribution is larger than that of carry and pass. To better illustrate this point, **we have added 5 visualizations of the distributions of shots, carry and pass in appendix C.3**. Shots are more risk-seeking, but defenseman performs shot less frequently, so they have a lower ranking based on the risk-seeking estimation.
> > >
> > >
> > > - "*Does this reveal that their EV is just so high that they are included regardless of risk preferences? I feel that there is a real missed opportunity to explain what the metric is doing in terms that could be useful to a practitioner.*"
> > >
> > > Connor McDavid appears in our top-10 risk-averse ranking (Table 2), but he is **not** included in the risk-seeking one (Table 1). Tyler Seguin, on the other hand, is highly ranked by both of our rankings, although his RiGIM values are significantly different (13.71 v.s., 1.78) and the rating has changed (6th v.s., 9th). Risk preference does influence evaluation, but this influence has not dominated the ranking. The practitioner can use our model to analyze a player's performance by observing how his or her rating is influenced by risk levels.
> > >
> > > - "*I think the missing explanation is that I'm having a hard time reasoning about what a risk-averse but highly successful action should look like. Everything that is positive seems to qualify as "risk-seeking" to some extent. Is there some way that this metric might lead to a more nuanced understanding of defensive behavior (which don't seem to have as strong traditional metrics associated with them).*"
> > >
> > > A general intuition we receive from our experiment and the sports experts is "stronger teams or players take more risks" (we introduce in our summary of update), but we believe the intuition must depend on the game context.
> > > Let's use the shots in Figure 1 as an example, where shot (b) is more risk-seeking than shot (a) (since the estimated value distribution of action (b) has larger uncertainty, please check the updated version of our paper). In general, shot (b) is preferred since its distribution has a mode on the high scoring chance (0.8). However, in some cases, when the game is tied and about to end, players might prefer (a) since they cannot afford the loss of next-goal scoring chance (which indicates their opponent will have a higher chance of scoring the next goal, see Figure 3).
> > >
> > > - *"Why would it be biased? There is no reason to think that the conventional metrics represent a risk-neutral measure of play. If you tune the hyper-parameter on a validation set that is unlikely to introduce bias."*
> > >
> > > We are aware of your concerns and will add the results after hyper-parameters fine-tuning (The experiments are still running on our machine, and we will update the results as soon as possible.) .

---

### Official Review · Reviewer_tcWQ · 2022-07-29

**Rating:** 3
**Confidence:** 4
**Soundness:** 3 good
**Presentation:** 2 fair
**Contribution:** 2 fair

**Summary:**

This is a new review, since my previous review was written for the wrong paper. My apologies for the mixup.

The paper introduces a methods based on RL for player evaluation in ice hockey and soccer. The method is called RIGIM
Distributional methods previously used in Atari and Mujoco are now used to evaluate real games.
The main contribution is the introduction of the RIGIM method. The description is quite technical and does not provide intuition how it works, what the short comings are, and how it can be improved.
An evaluation is provided, and a comparison to other methods.  The method has a low correlation to standard measures, but appears to perform somewhat better than other methods.
The paper does not really provide a discussion on the reasons for this.
My main problems with the paper are the clarity of the message and the clarity of the analysis.
It contains a description of a lot of work done, but what is the problem that is addressed? What is the contribution to the field? Why is it important?

Thanks very much for the message and for the request for clarification.

Thank you for the question. I can emphatize with the authors, and will elaborate on how the paper can be improved and why.

Examples: Section 4 is a technical section, that is missing an intuitive description of their method, and that is missing a high level explanation. Can the authors please add to rheir techjical description a high level explanation of the problems of related work that is adressed by their method. What is different, and why does it work better?

Examples of where the explanations can be improved; In Section 6.1 the authors write: “If we remove SP-CNF or replace it with other uncertainty estimators, most correlations become weaker except for the correlations with the SHP and SHG measures.” Can you please explain or speculate why this is the case?

Section 6, and Table 4 and 5 show correlations that are rather low, yet section 6.2 and section 6.3 fail to address this issue. The authors write: “RiGIM(c) becomes risk-seeking, and thus achieves a higher correlation with success measures. However, the correlations drop when c approaches 0. This observation is consistent with the fact that an overly risk-seeking estimate cannot reflect the real contributions of players.” Why is that so, why can it not reflect the real contributions?

The authors may also consider adding an explicit problem statement and revising their contributions to address questions of the field. As they are, the contributions are a list of work that has been done, failing to address the importance and which insights are contributed to the field.

Overall, I challenge the authors to let their descriptions transcend the description of what they have done, and provide explanations as to the why.

**Questions:**

Please provide a better explanation how the method works, how it builds on other work, how it compares to other work, what the strengths and weaknesses compared to other work are, and please provide better ideas on how to take this further.
The overall correlations to stnadard measures are quite low, why is that? What can be done to improve?

**Limitations:**

No, the limitations are listed very briefly, without real substantiation.

**Strengths And Weaknesses:**

Strengths:
new method for player evaluation
The method can learn from offline data
Use of distributional RL is interesting

Weaknesses:
Performance correlates poorly with standard measures, although somehat better than other methods
No real insight provided how the method works, and how it can be improved

---

> ### Author Response · Authors · 2022-08-02
> **Author response to comments from reviewer tcWQ - part 1**
>
> We thank the reviewer again for the prompt update and clarification of the review. We appreciate the reviewer for raising some points that require more explanations and clarification. We have significantly improved our paper as per your feedback. We hope these updates can more or less resolve your concerns.
>
> - "*Section 4 is a technical section, that is missing an intuitive description of their method, and that is missing a high-level explanation. Can the authors please add to their technical description a high-level explanation of the problems of related work that is addressed by their method? What is different, and why does it work better?*"
>
> **Response:** Section 4 introduces a detailed implementation of the "Uncertainty-Aware RL framework" (introduced in Section 3), where we show the risk-sensitive player evaluation requests modeling both the epistemic and aleatoric uncertainty inherent to the environment dynamic. Ignoring any of these uncertainties will cause inaccurate estimation, which we show in the experiment (by comparing with GIM and Na-RiGIM). In Section 4, we introduce a distributional-RL model and a feature-space density estimator to estimate the aforementioned epistemic and aleatoric uncertainties. To the best of our knowledge, none of the previous RL works are based on the direct estimation of both uncertainties. Since estimating the epistemic and aleatoric uncertainty together is very challenging in practice, the technical details are included to demonstrate why our model (distributional RL + SP-CNF) is a proper uncertainty estimator from both an intuitive and theoretical perspectives. We have added the clarification to our revised version (see lines 129-130 and lines 170-172).
>
> - "*Examples of where the explanations can be improved; In Section 6.1 the authors write: “If we remove SP-CNF or replace it with other uncertainty estimators, most correlations become weaker except for the correlations with the SHP and SHG measures.” Can you please explain or speculate why this is the case?*"
>
> **Response:** This is because SHP and SHG rarely happen in a season (since scoring with fewer players on ice is difficult). Since SP-CNF is a density estimator, it assigns a small density to these rarely-occurring events. According to equation (5), the events with a small density ($p(\cdot|{z}_{E})$) will be filtered. This filtering is necessary since events with negligible probability are considered to have large epistemic uncertainty (see Section 4.2), but the filtering sometimes causes a loss of information. This is the main reason why RiGIM does not have a leading correlation with SHP and SHG. Capturing the correct values for these out-of-distribution events is generally difficult (check [Gal2016]). We have expanded the explanation in the revised version (lines 297-301).
>
> [Gal2016] Gal, Yarin. "Uncertainty in deep learning." Ph.D. thesis, 2016.
>
> - "*Section 6, and Table 4 and 5 show correlations that are rather low, yet section 6.2 and section 6.3 fail to address this issue.*"
>
> **Response:** We have measured the correlations with both **success** measures and **penalty** measures, i.e., the correlations in the columns on the right-hand side of the dashed line (in Tables 4 and 5) should be as low as possible since the metrics refer to penalties.  In the terms of the scale of correlations, we study the correlation to **all** measures (including some penalty measures) instead of one. Having perfect correlations or anti-correlations to all these measures is impossible since they measure different aspects of a player. We on the other hand study which player evaluation metric can better correlate with these measures and thus forms a comprehensive evaluation of a player's overall performance.  We have clarified it in the revised version (see lines 281-285).
>
> - " *The authors write: “RiGIM(c) becomes risk-seeking, and thus achieves a higher correlation with success measures. However, the correlations drop when c approaches 0. This observation is consistent with the fact that an overly risk-seeking estimate cannot reflect the real contributions of players.” Why is that so, why can it not reflect the real contributions?*"
>
> **Response:** This is because when c approaches 0, RiGIM will focus on the quantile level 1 (since 1-c=1, check the line after equation (5)), which corresponds to the largest value in the support of a distribution. In our case, this is the most optimistic estimation of the value that an action can achieve. For example, in the value distribution of shots, $Z^{1}(\cdot, shot)$ will be close to 1 most of the time, since the best outcome of a shot is scoring. However, in fact, only a few shots can turn into goals for both soccer and ice hockey. The overly risk-seeking estimation can induce a mismatch between estimated values and game facts. We have expanded the explanation in the revised version (see lines 320-323).

---

> > ### Author Response · Authors · 2022-08-02
> > **Author response to comments from reviewer tcWQ - part 2**
> >
> > - "*The authors may also consider adding an explicit problem statement and revising their contributions to address questions of the field...*"
> >
> >  **Response:**  Our work involves interdisciplinary study from both sports analytics and Reinforcement Learning (RL). From the perspective of sports analytics, player evaluation is an important topic that has been studied by many previous works(see our related work Section 2). Assigning values to players' actions is a common approach for player evaluation, but none of the previous works has considered risk or uncertainty during evaluation. Our work extends the action-value approach by adding the dimension of risk to evaluation. This extension is **fundamental** since sports games (especially team sports) have **inherent risk and uncertainty**, which should not be ignored during evaluation. When it comes to RL, some recent works (i.e., offline RL) have studied the approach of adjusting action values according to uncertainty estimates, but these works often **do not explicitly model the epistemic and aleatoric uncertainties** (in most cases, the estimation can not be disentangled into epistemic and aleatoric uncertainties, see [Mavrin2019]). Our work proposes a framework that enables this estimation. We have clarified it in the revised version (see lines 55-63).
> >
> > [Mavrin2019] Borislav Mavrin, Hengshuai Yao, Linglong Kong, Kaiwen Wu, and Yaoliang Yu. Distributional reinforcement learning for efficient exploration. In International Conference on Machine Learning (ICML), volume 97, pages 4424–4434, 2019.
> >
> > - "*How it builds on other work, how it compares to other work, what the strengths and weaknesses compared to other work are,*"
> >
> > **Response:** Our empirical evaluation follows an ablation design: We iteratively remove parts from RiGIM, and we show these simplifications degrade RiGIM to risk-neural or tabular-based baselines. This ablation study allows us to analyze the influence of each component in RiGIM, including the risk-sensitivity and uncertainty estimation models. The results in sections 6.1-6.3 are presented by following the structure of the ablation study. We have expanded the explanation of experiment results, including the strengths and weaknesses compared to comparison methods in the revised version.
> >
> > - "*The method is called RIGIM Distributional methods previously used in Atari and Mujoco are now used to evaluate real games.*"
> >
> > **Response:** Our method RIGIM is new and proposed for agent evaluation. It is novel and has not been applied to solve the control problems in Atari and Mujoco.
> >
> > - "*What is the problem that is addressed?*"
> >
> > **Response:** We explain the problem in the first paragraph (from lines 17 to 23) of the introduction. The last sentence of this paragraph explains that the paper tackles the problem of player evaluation. To be more specific, we are considering a problem of assigning proper credits to players' actions by conditioning on a risk level. This is a fundamental challenge in sports analytics.

---

> > > ### Comment · Reviewer_tcWQ · 2022-08-08
> > > **Rebuttal**
> > >
> > > Thanks very much for the amended version of the paper, and for the thoughtful and important clarifications and answers to the questions. Indeed, this does provide more insight. Many questions were asked, many answers were provided.
> > >
> > > After careful reading I do not feel that the work has achieved the level of maturity, substance, and clarity for this  conference, and I have chosen not to change my assessment.

---

### Review · Ethics_Reviewer_rBMw · 2022-08-23

**Recommendation:**

There are no outstanding ethical issues to address in this paper. A discussion of real-world impact if a metric like this were to be implemented would strengthen the paper, but in its current form, that is not necessary.

**Ethics Review:**

The paper describes a risk-aware player evaluation metric, leveraging RF-specific ideas to quantify aleotoric and epistemic uncertainty. The paper was originally flagged by reviewer mdqR because of the involvement on human subjects, aka players here.

Because the work deals primarily with metric creation and isn't actively working with human subjects, there are no ethical concerns with this paper. Real-world implementation might distort player incentives and playing styles, and this may create both technical and sociological wrinkles in the current formulation. A discussion of that would strengthen the paper, but in its current form, that is not necessary.

---

### Author Response · Authors · 2022-08-01
**Summary of Updates**

Dear Reviewers, Area Chairs, and Program Chairs,

We are greatly thankful for the insightful comments and suggestions, which are very helpful for us to further improve this work.
The major concern is over our experiment section. We agree some clarifications, explanations, and experiments should be added (**highlighted in blue**). To clarify our modifications and prevent misunderstanding, we summarize our major updates in the following:

- **Player Ranking.** We expand section 5.2 by providing more explanations for the risk-sensitive ranking results (Table 1 and 2). We show the motivation of rankings under different confidence levels $c$ and a brief explanation of the results from perspective of action frequency.

- **Confidence level.** We show the motivation of applying $c$ as a hyper-parameters (Section 6.1) and briefly explain why we set $c$ to some specific values in different empirical studies.

- **Correlation with Success Measures.** We add the motivations of applying the correlations with success measures to evaluate the player ranking metrics (Section 6.1). We also expand the explanation of our experiment results according to the comments from our reviewers.

- **Significance Test.** In order to show our results are significantly different from comparison methods, we add a Wilcoxon signed rank test to the experiment results in Section 6.3.

- **Experiment Results.** We have significantly expanded the discussion about experiment results by carefully analyzing the phenomenon we have observed.

- **Clarification about the main motivation and contribution.** We have revised our contribution section by following a clear structure from our uncertainty-aware RL framework. We present our contributions to both RL and sports analytics according to reviewers' comments.

Apart from the academic contributions in the paper, our risk-measuring method has **realistic contribution**: we presented our ideas to some experts from the sports industry. They agreed on the value of computing the risk of player movements and suggested ranking players or teams according to their risk. Their intuition is **stronger teams or players take more risks**. This intuition is consistent with our findings in the risk-sensitive experiment (Section 6.2).
Sports, as an important part of the entertainment industry, often admires risk.
On the other hand,  most RL algorithms (typically offline RL) prefer a **conservative policy** to handle risk and uncertainty. We believe this difference can inspire many future studies.

---

> ### Author Response · Authors · 2022-08-09
> **More Updates**
>
> Dear Reviewers, Area Chairs, and Program Chairs,
>
> Thanks for the discussing paper with us. Based on the discussion, we add more updates to our paper. Hope they can resolve your concerns:
>
> 1. As reviewer kCNJ suggested, we have explored the option of determining the risk level by utilizing the validation dataset. We have added the results to Section 6.1. (Tables 4 and 5)
>
> 2. In order to resolve the concerns (why defensemen are risk averse) from reviewer kCNJ, we have added 5 visualizations of the distributions of shots, carry and pass in Appendix C.3.
>
> 3. As reviewer mdqR suggested, we have presented the computational complexity and model complexity in Appendix A.4.
>
> 4. We have studied the scale of epistemic uncertainty as more games are observed and reported the results in Appendix C.4 according to the comments from reviewer mdqR.

---

### Meta-Review · Area_Chair_hb1s · 2022-08-26

**Recommendation:** Accept
**Confidence:** Less certain

**Metareview:**

The reviews on this paper were mixed. One of the reviewers raised several concerns about the work; although they were valid concerns, our assessment is that the authors have addressed them to a satisfactory degree in their rebuttal.

This is a solid paper describing a systematic, careful, application of techniques from supervised and reinforcement learning to the problem of evaluating sport players taking risk into account. In preparing their final manuscript, we suggest the authors make an effort to explain the application as well as possible to the NeurIPS audience, who may not be familiar with player evaluation in sports game. The interesting discussion between authors and reviewers that followed the initial reviews can be a good source of the type of questions the community  might be interested in. Although the authors have already modified the paper in response to this discussion, we suggest they try to incorporate as much of it as possible to the final version of the paper.

**Award:**

No

---

### Decision · Program_Chairs · 2022-09-14

Accept